# The Molecular Weight Distribution of Occluded Hydrocarbon Gases in the Khibiny Nepheline–Syenite Massif (Kola Peninsula, NW Russia) in Relation to the Problem of Their Origin

**Valentin A. Nivin, Vyacheslav V. Pukha \*, Olga D. Mokrushina and Julia A. Mikhailova**

Geological Institute of the Kola Science Centre, Russian Academy of Sciences, 14 Fersman Street, 184209 Apatity, Russia

\* Correspondence: puhanot@gmail.com

**Abstract:** The origin of hydrogen–hydrocarbon gases present in the rocks of the Khibiny massif in unusually high concentrations has been the subject of many years of discussion. To assess the role of potential mechanisms and relative time of formation of gases occluded in inclusions in minerals, the molecular weight distribution of $C_1$–$C_5$ alkanes in the main rock types of the Khibiny massif was studied. For this purpose, the occluded gases were extracted from rocks by mechanical grinding and their composition was analyzed on a gas chromatograph. It is established that the molecular weight distribution of occluded hydrocarbon gases in the Khibiny massif corresponds to the classical Anderson–Schulz–Flory distribution. In addition, the slopes of the linear relationships are relatively steep. This indicates a predominantly abiogenic origin of the occluded gases of the Khibiny massif. At the same time, a small proportion of biogenic hydrocarbons is present and is associated with the influence of meteoric waters. It was also found that in the Khibiny massif, the proportion of relatively high-temperature gases decreases towards the Main foidolite Ring in the following sequence: foyaite and khibinite–trachytoid khibinite–rischorrite and lyavochorrite–foidolites and apatite–nepheline ores. In the same sequence, an increase in the proportion of heavy hydrocarbons and the increasing role of oxidation and condensation reactions in the transformation of hydrocarbons occurs.

**Keywords:** abiogenic hydrocarbon gases; occluded gases; molecular weight distribution; hydrogen; nepheline syenites; foidolites; fluid inclusions

## 1. Introduction

Many peralkaline (with molar ratio (Na + K)/Al > 1) intrusive rocks are significantly enriched in the hydrogen and hydrocarbon gases (mainly methane) of putative abiogenic origin. This has been well documented in the Ilímaussaq (Greenland) [1], Strange Lake (Canada) [2], Lovozero and Khibiny (Kola Peninsula, Russia) [3,4] plutons, with the two latter neighboring massifs being particularly enriched in these gases. Usually hydrogen–hydrocarbon gases are enclosed in vacuoles of fluid inclusions in rock-forming and accessory minerals (co-called 'occluded gases'). In the Khibiny and Lovozero massifs, other forms of occurrence (morphological types) of hydrogen–hydrocarbon gases are also known. These are 'free gases' and 'diffusely dispersed gases' [4–7]. Free gases fill systems of interconnected (micro)fractures, as well as other cavities in rocks. Diffusely dispersed gases fill closed and semi-permeable thin and subcapillary cracks and remain mostly in an adsorbed state; their movement is dominated by a diffusive transfer.

The issues of the origin and emission of reduced hydrogen–hydrocarbon gases are of both scientific and practical interest. These are, in particular, the role of reduced fluids

in the transfer and concentration of ore elements and formation of mineral deposits [8–10], the abiogenic synthesis of organic molecules, the development of the deep biosphere, the origin of life on Earth [11–13], the provision of sustainable energy supply at low ecological and economic costs [11,12,14], the safe mining of ore deposits [4,5,15], the creation of models of the carbon cycle and degassing of the Earth [11,16], the universal theory of petroleum genesis [11,17], and the assessment of the scale of the lithospheric greenhouse and ozone-depleting gases runoff into the atmosphere [13,16,18].

Despite more than a half-century history of studies of hydrogen–hydrocarbon gases in rocks of nepheline–syenite complexes, the mechanism, conditions, and relative time of their formation are still the subject of discussion [5,7,19–35]. Previously proposed hypotheses for the genesis of hydrocarbon gases are based on variations in their concentrations in rocks and minerals, the isotopic composition of carbon and hydrogen, thermodynamic calculations, and thermobarometry of fluid microinclusions. These hypotheses are as follows:

(1) A direct mantle origin [21,22,29]. Conclusions about the mantle source of methane based on their C-D isotope systematics. Sub-solidus abiogenic $CH_4$-oxidation ($4CH_4 + O_2 \rightarrow 2C_2H_6 + 2H_2O$) and polymerization ($nCH_4 \rightarrow C_nH_{2n+2} + (n-1)\ H_2$) reactions produced higher hydrocarbons.

(2) A late-magmatic (below 600 °C) origin by re-speciation of a C-O-H fluid [36,37]. The speciation of various fluids in the C-O-H system is influenced by changing temperature, pressure, oxygen fugacity, and graphite activity. A change in these parameters leads to a change in the composition of the fluid.

(3) A post-magmatic (350–400 °C) origin by Fischer–Tropsch types of reactions between an exsolved magmatic $CO_2$-dominant fluid and $H_2$ produced from hydrothermal mineralogical reactions [2,30]. Fischer–Tropsch synthesis involves a series of step reactions, which can be represented by equations such as:

$$nCO + (2n+1)\ H_2 \rightarrow C_nH_{2n+2} + nH_2O,\ nCO_2 + (3n+1)\ H_2 \rightarrow C_nH_{2n+2} + 2nH_2O. \qquad (1)$$

These reactions are catalyzed in the presence of group VIII metals, Fe-oxides, Fe-silicates, and hydrated silicates.

(4) A mixed magmatic/thermogenic origin [32]. According to this hypothesis, magmatically derived abiogenic hydrocarbons may have mixed with biogenic hydrocarbons derived from the surrounding country rocks.

(5) A thermogenic origin [31]. According to this theory, hydrocarbons found throughout peralkaline complexes are the result of the migration of external thermogenically-derived hydrocarbon fluids into these complexes.

To establish the mechanism and conditions for the formation of hydrocarbon gases in plutonic rocks, researchers used the various ratios of the concentrations of these gases, for example, the molecular weight distribution [2,13,38–40]. Particularly, in Fischer–Tropsch synthesis products, the molecular ratios of hydrocarbons with successive carbon numbers are constant ($C_2/C_1 \approx C_3/C_2 \approx C_{n+1}/C_n$), resulting in the Anderson–Schulz–Flory distribution of hydrocarbons [41,42]. Therefore, the plot of log $X_i$ versus $C_n$ (where X is concentration) should give a straight line [42]. An accordance with the classical Anderson–Schulz–Flory distribution and a relatively steep slope of the linear dependence is considered as a criterion of abiogenic origin of natural gases.

This article discusses the features of the molecular weight distribution of hydrocarbons occluded in minerals from all main rock types of the Khibiny massif. For this purpose, chromatographic analyses of occluded gases accumulated over many years, including by the authors, were collected, systematized, and revised.

## 2. Geological and Occluded Gas Geochemical Backgrounds

### 2.1. Khibiny Massif

The Khibiny massif is located in the southwest of the Kola Peninsula at the contact of Archean gneisses and Proterozoic sedimentary-volcanogenic rocks and covers an area of 1327 km² [43–46]. The age of the massif is 360–380 Ma [47]. The Khibiny massif has a concentrically zoned structure (Figure 1). In plain view, the massif is elliptical (45 × 35 km) and vertically it is cone-like, with its apex pointing downward. The massif consists dominantly of nepheline syenites (about 90% of the outcrop area) and foidolites (8% of the outcrop area) that intrude into the nepheline syenites along the two cone-like faults: the Main Ring (or Main foidolite Ring) and Minor Ring faults [48]. The foidolites of the Main Ring accommodate all the apatite deposits and occurrences. The apatite–nepheline and titanite–apatite–nepheline ores form stockworks in the apical parts of the foidolite intrusions [49].

In the Khibiny massif, several varieties of nepheline syenites are distinguished, which have historically accepted local names [50] that are important for describing the geology, and we will use them in the text below. These rock names are as follows:

- *khibinite* is a eudialyte-bearing nepheline syenite with aegirine, alkali amphibole, and many accessory minerals, particularly those containing Ti and Zr;
- *foyaite* is a massive, less often weakly trachytoid, leucocratic nepheline syenite;
- *rischorrite* is a leucocratic nepheline syenite in which the nepheline crystals are poikilitically enclosed in microcline perthite;
- *lyavochorrite* is a leucocratic nepheline syenite in which only part of the feldspar crystals is poikilitic.

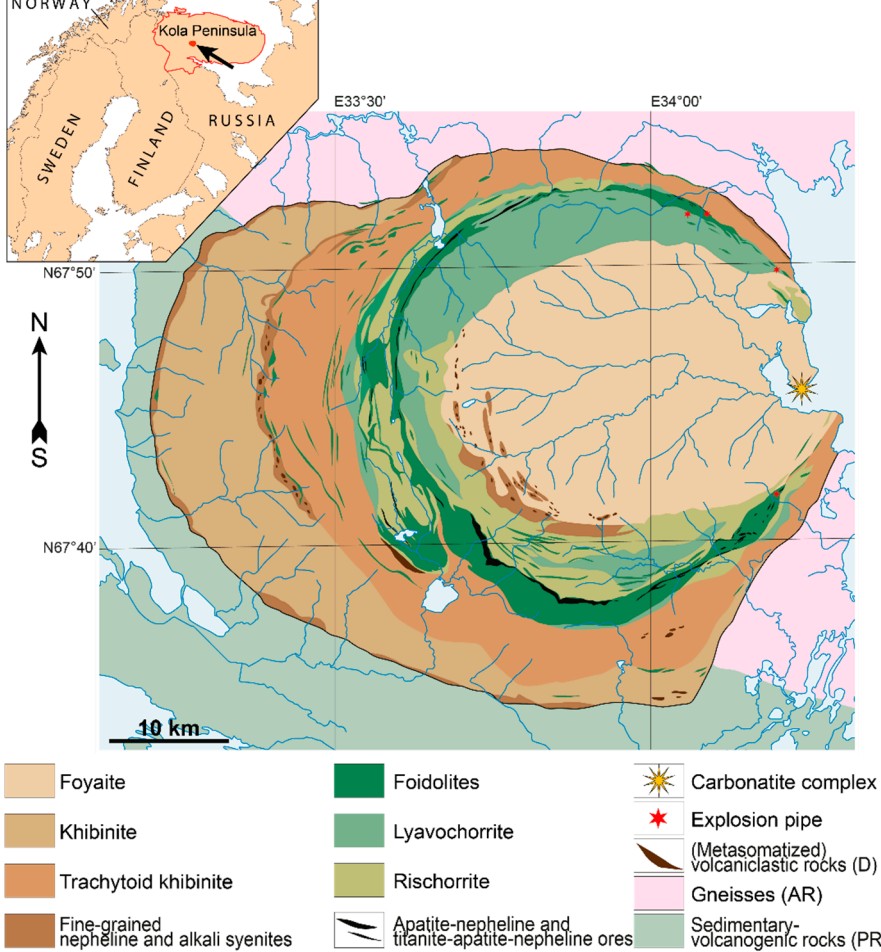

**Figure 1.** Geological scheme of the Khibiny massif [43].

Khibinite and trachytoid khibinite are located outside the Main Ring, foyaite is situated inside the Main Ring, and rischorrite and lyavochorrite are located on both sides (Figure 1). Most of the pegmatites and hydrothermal veins are at the contact between rischorrite and foyaite. The carbonatite complex is located near the eastern contact of the massif [51]. The core of this complex, about one kilometer across, is composed of albite-carbonate, biotite–carbonate, aegirine–carbonate–biotite rocks, and carbonatites, and is surrounded by stockworks of calcite–albite rocks, carbonatites with Ba–REE–Sr mineralization, phoscorites, and carbonate–zeolite rocks. The carbonatization of surrounded foyaite is observed.

The Khibiny massif is surrounded by a halo of fenites up to 500 m wide [52]. In the central parts of the massif there are numerous roof xenoliths, both unaltered (basalt, basalt tuff, tuffite) and intensely metasomatized [53].

The Main foidolite Ring is the "axis of symmetry" of the Khibiny massif. The mineral, modal, and chemical compositions of rocks, the chemical compositions of rock-forming minerals, change symmetrically with respect to the Main Ring. For example, in the composition of rock-forming clinopyroxenes, when approaching the Main Ring, the content of diopside end-member increases [54], and in amphiboles, the concentration of potassium and magnesium increases [55,56].

*2.2. Occluded Gases*

Unusually high concentrations of hydrocarbon gases for igneous rocks were first discovered during the mining of the Khibiny apatite–nepheline deposits [57]. Since then, dozens of works, including monographs [3,4,58], have been devoted to the study of various forms of occurrence of these gases in the Khibiny massif alone. The majority of attention was paid to the gases occluded in fluid inclusions in the minerals, since gases of this morphological type are better available for research than others. The following approaches to the study of occluded gases were used: (1) the study of individual fluid inclusions in minerals and (2) the study of the bulk composition of occluded gases in the rock as a whole. In the latter case, gases were extracted by mechanical grinding of the rock sample and analyzed on a gas chromatograph (this technique is described in detail below).

According to the results of gas extraction by mechanical grinding of samples and gas chromatographic analysis [5,59], the main component of the Khibiny occluded gases is methane. Its concentration is 70–90 vol.% of occluded gases, and the content in the rocks reaches 365 $cm^3$/kg with an average (median) value of about 10 $cm^3$/kg. Molecular hydrogen, methane homologues (up to and including pentanes), $N_2$ and $O_2$, alkenes, and helium are constantly present in subordinate and microquantities. With a decrease in the content of methane, the concentration of non-hydrocarbon gases increases. Carbon dioxide and carbon monoxide are quite rare. The distribution of occluded gases in the rocks of the Khibiny massif is irregular [5,59]. Sometimes, even in the rocks of the same type, without visible macroscopic differences, at a distance of less than one meter, concentrations of gases can differ by two orders of magnitude. Maximum variations in the composition and content of gases are observed in the rocks of the Main Ring. Among the main types of rocks, the most saturated in occluded gases are lyavochorrite and urtite, while apatite–nepheline ores are characterized by low gas content.

The hydrogen–hydrocarbon and/or substantially hydrocarbon composition of occluded gases extracted by grinding the samples is consistent with the microthermobarometry and Raman spectroscopy data obtained from individual inclusions [32,60], as well as experiments on gas extraction by dissolving samples in acids [4].

The most important carriers of fluid inclusions are rock-forming nepheline and alkali feldspar, and sodalite, analcime, eudialyte-group minerals, titaniferous magnetite, and aenigmatite. The gas saturation of minerals is directly related to the total gas content in the rock [27], i.e., the more occluded gases in rock's minerals, the more free and diffusely dispersed gases in this rock. Being practically the only or at least the main source of oc-

cluded gases, fluid inclusions in Khibiny minerals at room temperature are predominantly single-phase (gas) or, less often, two- and multi-phase, and have a rounded, tubular, or irregular shape [3,32,60] (Figure 2). Some inclusions are in the form of negative crystals. The predominant size of the fluid inclusions is <15 μm, but in rare cases they reach 150 μm. Most of the observed fluid inclusions are secondary, localized in planar zones that cross-cut the host crystal in different directions. Much less common are primary and primary–secondary fluid inclusions, both single and in small groups, which mark the crystal growth zones. There are also melt inclusions with a gas bubble [32,61].

Methane predominates in the composition of individual fluid inclusions; there are also fluid inclusions consisting of an aqueous solution and inclusions containing methane and water. In ultraviolet light, fluorescent rims are observed in some inclusions, indicating the presence of liquid hydrocarbons, in particular (according to Raman spectroscopy), $C_6H_{14}$ [62] and $C_7H_{16}$ [63]. Molecular hydrogen and nitrogen are also identified by the Raman spectroscopy method in the composition of gas inclusions [32,64]. Daughter solid phases (halite, nahcolite) are rare in aqueous inclusions. Carbon dioxide and carbon dioxide–water inclusions were found only in carbonatites.

The hydrocarbon inclusions are homogenized into liquid or vapor in the temperature range of −62 °C to −119 °C. The prevailing homogenization temperatures of −80−−84 °C indicate a significantly methane composition of the gases. There were no signs of $CO_2$ presence. The homogenization of $H_2O$ inclusions, depending on the liquid–vapor ratio varying from 10 to 80%, occurs at temperatures of 109–350 °C (less steam—lower temperature), but in most cases at 270–350 °C. According to microthermobarometry data, fluids were trapped in inclusions at temperatures of 350 °C and below and pressures of 0.2–2.1 kbar. In addition, the most high-density, almost pure methane inclusions formed earlier and at relatively high pressures.

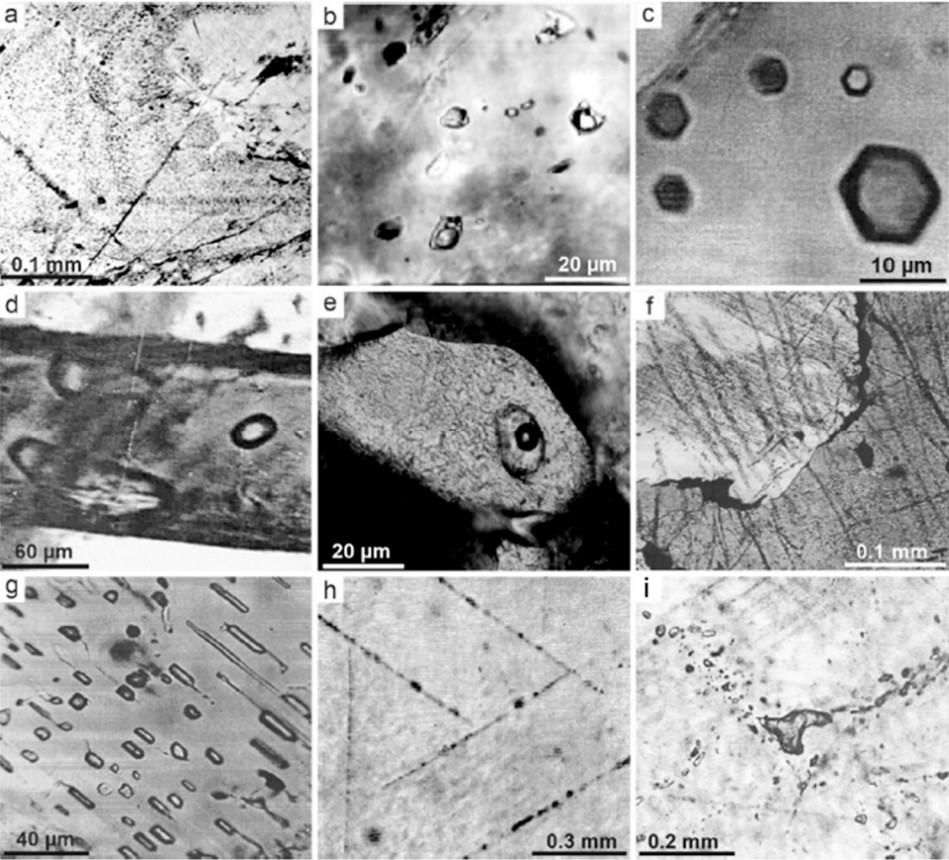

**Figure 2.** Primary (**a**–**e**) and secondary (**f**–**i**) fluid inclusions in the Khibiny minerals: (**a**) gas (hydrocarbon-bearing) inclusions along growth zones; (**b**) two-phase fluid inclusions in the nepheline [62];

gas inclusions in nepheline (**c**) and aegirine (**d**) of the rischorrite; (**e**) gas–liquid inclusion in titanite from the apatite–nepheline ore; (**f,g**) hydrocarbon-bearing fluid (mainly gaseous) inclusions in the nepheline of urtite; (**h,i**) fluid (gas and gas–liquid) hydrocarbon-bearing inclusions in the eudialyte-group mineral [3].

## 3. Materials and Methods

For this study, 238 samples of the main rock types of the Khibiny massif were taken, in which the full spectrum of gaseous ($C_1$–$C_5$) alkanes was determined. The samples included both igneous rocks such as khibinite and foyaite, as well as pegmatites and hydrothermal veins.

Measurements of the compositions of occluded gases were carried out in different years, but by the same method. For the study of occluded gases, the method of mechanical extraction of gases was used, followed by analysis on a gas chromatograph. This technique makes it possible to determine the gross composition of gases, since both primary and secondary inclusions are opened during rock grinding. Compared to the thermal gross gas extraction method, the mechanical extraction method reduces the risk of new gas generation. However, with the thermal method, the formation of gases can occur as a result of various reactions when the rock is heated to high temperatures.

Before crushing, the samples were washed to remove any organic matter. Depending on the sample size, two methods were applied to extract the occluded gases [4]. The crushing of large (200–350 g) samples was carried out for 20 min on a vibrating mill in special pre-vacuum sealed stainless steel beakers with grinding balls. With a fraction of −10 + 1 mm loaded into the beaker, about 60–70% of the sample was crushed to particles of 0.05 mm. After the sample was crushed, the released gas was pumped out on a mercury device and then injected into the chromatograph using a syringe.

For small (0.5–1.0 g) samples, a vibrating chamber of small volume (about 2.5 cm³) built into the gas system of the chromatograph was used, in which the sample fraction of −0.63 + 0.25 mm, together with grinding bodies, was loaded. The grinding was carried out for 20 min in a helium atmosphere at room temperature and pressure. The released gases, after switching the dosing valve, were displaced by the carrier gas into the chromatographic column.

Special experiments were carried out to assess the possibility of new formation of gas components during the grinding of samples containing dispersed organic matter. Grinding by both methods of samples before and after chloroform and alcohol-benzene extraction of organics and chromatographic analysis showed that the presence of organic matter does not affect the composition of the released occluded gases. Other experiments suggest the release of an insignificant amount of hydrogen from the material of the mills (steel) during the grinding of samples. However, this volume can be neglected at the revealed levels of natural $H_2$ concentration in the studied samples.

Gas analysis was carried out on chromatographs PE F-30, TSVET-102, and TSVET-104. Helium and argon were used as carrier gases. For calibration, sets of standard gas mixtures were used, covering the entire range of possible concentrations of measured gases in the studied rocks. The minimum detectable concentrations of individual gases are 0.0005 vol.% for $CH_4$, 0.00005 vol.% for $C_2H_6$, 0.00032 vol.% for $H_2$, 0.00045 vol.% for He, 0.013 vol.% for CO, and 0.042 vol.% for $CO_2$. The standard deviation of the individual components was 0.4–0.8 vol.% and the coefficient of variation ranged from 2.7 to 4.6%.

The results of the analysis of occluded gases in the rocks of Khibiny massif according to the described above method have been repeatedly confirmed by other laboratories [65,66].

## 4. Results

Table 1 presents data on the composition of occluded hydrocarbon gases in different types of rocks of the Khibiny massif, and Table 2 shows data on the composition of hydrogen, carbon dioxide, nitrogen, and oxygen.

**Table 1.** Concentrations of occluded hydrocarbon gases in the Khibiny massif {min-max/median (number of analyses)}, $cm^3$/kg.

| Rock | $CH_4$ | $C_2H_6$ | $C_3H_8$ | $iC_4H_{10}$ | $nC_4H_{10}$ | $iC_5H_{12}$ | $nC_5H_{12}$ |
|---|---|---|---|---|---|---|---|
| Khibinite | 4.29–77.0 | 0.09–3.10 | 0.005–0.312 | 0.0002–0.0230 | 0.0007–0.0620 | 0.0001–0.0167 | 0.00005–0.0028 |
| | 25.8 (19) | 0.51 (19) | 0.051 (19) | 0.0028 (19) | 0.0083 (19) | 0.0012 (19) | 0.00043 (19) |
| Trachytoid khibinite | 1.04–66.6 | 0.03–3.20 | 0.002–0.358 | 0.0001–0.0320 | 0.0003–0.0670 | 0.0–0.010 | 0.00002–0.0043 |
| | 18.8 (27) | 0.62 (27) | 0.049 (26) | 0.0023 (27) | 0.0091 (27) | 0.0012 (27) | 0.00052 (27) |
| Rischorrite | 2.9–116.5 | 0.12–6.92 | 0.008–0.770 | 0.0005–0.060 | 0.0013–0.1450 | 0.0003–0.0470 | 0.00010–0.0180 |
| | 9.6 (15) | 0.52 (15) | 0.049 (15) | 0.0025 (15) | 0.0088 (15) | 0.0014 (15) | 0.00069 (15) |
| Ijolite | 1.0–74.60 | 0.04–2.52 | 0.006–0.540 | 0.0002–0.10 | 0.0009–0.120 | 0.0001–0.0470 | 0.00002–0.0480 |
| | 12.4 (27) | 0.55 (27) | 0.071 (27) | 0.0077 (27) | 0.0110 (27) | 0.0030 (27) | 0.00130 (27) |
| Urtite | 4.77–86.0 | 0.12–5.12 | 0.012–0.520 | 0.0009–0.0690 | 0.0023–0.0930 | 0.0003–0.0330 | 0.00015–0.0210 |
| | 23.4 (60) | 0.98 (60) | 0.10 (60) | 0.0083 (60) | 0.0209 (60) | 0.0043 (60) | 0.00190 (60) |
| Apatite–nepheline ore | 0.05–10.8 | 0.01–1.04 | 0.001–0.075 | 0.0001–0.0076 | 0.0001–0.0150 | 0.0–0.0041 | 0.00001–0.0014 |
| | 2.4 (9) | 0.12 (9) | 0.011 (9) | 0.0007 (9) | 0.0015 (9) | 0.0006 (9) | 0.00016 (9) |
| Lyavochorrite | 6.15–74.5 | 0.23–2.96 | 0.011–0.310 | 0.0007–0.0260 | 0.0020–0.0571 | 0.0003–0.0174 | 0.00003–0.0053 |
| | 19.5 (18) | 1.19 (18) | 0.108 (18) | 0.0074 (18) | 0.0195 (18) | 0.0027 (18) | 0.00120 (17) |
| Foyaite | 2.97–33.2 | 0.05–1.12 | 0.002–0.078 | 0.0001–0.0054 | 0.0002–0.0110 | 0.0–0.0021 | 0.00002–0.0009 |
| | 9.3 (13) | 0.20 (13) | 0.010 (13) | 0.0005 (13) | 0.0021 (13) | 0.0003 (13) | 0.00018 (13) |
| Carbonatized foyaite | 0.04–0.30 | 0.0–0.01 | 0.0–0.002 | 0.0–0.0012 | 0.0001–0.0005 | b.d.l. | b.d.l. |
| | 0.1 (4) | 0.0 (4) | 0.001 (4) | 0.0001 (4) | 0.0003 (4) | | |
| Carbonatites | 0.17–4.08 | 0.01–0.36 | 0.002–0.081 | 0.0004–0.0120 | 0.0003–0.0170 | 0.0001–0.0032 | 0.00003–0.0021 |
| | 0.8 (19) | 0.06 (19) | 0.014 (19) | 0.0024 (19) | 0.0026 (19) | 0.0008 (19) | 0.00050 (19) |
| Carbonate-albite and albite-carbonate rocks | 0.13–2.74 | 0.01–0.25 | 0.002–0.090 | 0.0003–0.0220 | 0.0003–0.0180 | 0.0001–0.0050 | 0.00009–0.0043 |
| | 1.1 (13) | 0.09 (13) | 0.045 (13) | 0.0130 (13) | 0.0064 (13) | 0.0032 (13) | 0.00160 (13) |
| Pegmatites | 9.29–31.6 | 0.32–2.31 | 0.089–0.170 | 0.0039–0.0130 | 0.0072–0.0280 | 0.0022–0.0046 | 0.00078–0.0014 |
| | 23.1 (3) | 0.84 (3) | 0.092 (3) | 0.0060 (3) | 0.0140 (3) | 0.0030 (3) | 0.0010 (3) |
| Hydrothermalites | 6.23–11.0 | 0.29–0.98 | 0.022–0.170 | 0.0014–0.0420 | 0.0026–0.0360 | 0.0001–0.0086 | 0.00011–0.0053 |
| | 8.6 (2) | 0.64 (2) | 0.096 (2) | 0.0217 (2) | 0.0193 (2) | 0.0043 (2) | 0.00271 (2) |
| Fenite | 0.01–13.4 | 0.0–0.12 | 0.0–0.005 | 0.0–0.0002 | 0.0–0.0006 | b.d.l. | b.d.l. |
| | 0.5 (7) | 0.01 (7) | 0.0 (7) | 0.0 (7) | 0.0 (7) | | |

b.d.l.—below detection limit.

**Table 2.** Concentrations of occluded gases in the Khibiny massif {min-max/median (number of analyses)}, $cm^3$/kg.

| Rock | $H_2$ | $CO_2$ | $N_2$ | $O_2$ |
|---|---|---|---|---|
| Khibinite | 0.37–2.30 | 0.04–0.31 | 0.19–1.70 | 0.013–0.46 |
| | 1.27 (19) | 0.17 (3) | 0.53 (18) | 0.07 (18) |
| Trachytoid khibinite | 0.61–1.89 | 0.21–0.21 | 0.07–2.60 | 0.025–0.45 |
| | 1.04 (27) | 0.21 (1) | 0.38 (27) | 0.05 (19) |
| Rischorrite | 0.38–5.10 | 0.34–3.26 | 0.35–1.55 | 0.030–0.41 |
| | 1.39 (15) | 0.70 (5) | 0.79 (13) | 0.07 (13) |
| Ijolite | 0.17–26.4 | 0.07–1.24 | 0.25–11.3 | 0.020–0.55 |
| | 0.67 (26) | 0.68 (7) | 0.99 (17) | 0.12 (17) |
| Urtite | 0.19–5.10 | 0.01–7.46 | 0.49–9.53 | 0.025–0.84 |
| | 0.95 (60) | 0.55 (9) | 1.26 (50) | 0.15 (50) |
| Apatite–nepheline ore | 0.12–2.83 | b.d.l. | 0.25–2.28 | 0.041–0.41 |
| | 0.41 (9) | | 0.75 (9) | 0.14 (9) |
| Lyavochorrite | 0.53–9.43 | 0.01–6.6 | 0.29–2.31 | 0.024–0.32 |
| | 1.24 (18) | 1.94 (4) | 0.98 (14) | 0.14 (14) |
| Foyaite | 0.58–3.80 | 0.17–1.75 | 0.16–1.98 | 0.007–0.21 |
| | 0.88 (13) | 0.96 (2) | 0.29 (10) | 0.02 (10) |
| Carbonatized foyaite | 0.58–2.51 | 1.49–20.0 | 0.30–1.0 | 0.021–0.06 |
| | 1.20 (4) | 9.19 (4) | 0.47 (4) | 0.03 (4) |
| Carbonatite | 0.04–14.4 | 0.33–16.20 | 0.19–1.62 | 0.003–0.21 |
| | 2.22 (19) | 3.61 (19) | 0.43 (19) | 0.04 (19) |

| Carbonate-albite and albite-carbonate rocks | 0.92–8.33 4.07 (13) | 0.99–14.20 4.35 (13) | 0.24–2.64 0.57 (13) | 0.015–0.32 0.05 (13) |
|---|---|---|---|---|
| Pegmatite | 0.54–1.19 1.19 (3) | b.d.l. | 0.57–2.65 1.49 (3) | 0.079–0.19 0.08 (3) |
| Hydrothermalite | 0.24–0.87 0.56 (2) | b.d.l. | 0.87–1.05 0.96 (2) | 0.083–0.35 0.22 (2) |
| Fenite | 0.18–4.59 0.59 (7) | 0.03–15.85 0.11 (4) | 0.33–2.21 0.93 (5) | 0.030–0.36 0.17 (4) |

b.d.l.—below detection limit.

Figures 3 and 4 show median molecular weight-distributions of hydrocarbon gases in the magmatic rocks and apatite–nepheline ore (Figure 3), as well as in the carbonatites, hydrothermalites, pegmatites, and fenite (Figure 4) of the Khibiny massif. Molecular weight distribution of alkanes in the Khibiny rocks, except for the carbonatized foyaite, is consistent with the classical Anderson–Schulz–Flory distribution model. The plot of log $X_i$ versus $C_n$ (where X is concentration) gives a straight line [42]. The slope of the graphs of the log-linear dependence varies significantly, decreasing from foyaite and khibinite to foidolites and apatite–nepheline ores (Figure 3). The slopes of the molecular weight-distribution plots for pegmatites and hydrothermal veins are similar to the slopes of the plots for foidolites and apatite–nepheline rocks (Figure 4). The graphs of the molecular mass distribution of alkanes from carbonatites and albite-carbonate rocks have minimal slopes. The graph for fenite has the sharpest slope. The average $C_{n+1}/C_n$ ratio ranges from 0.11 in foyaite to 0.25 in the rocks of the Carbonatite complex.

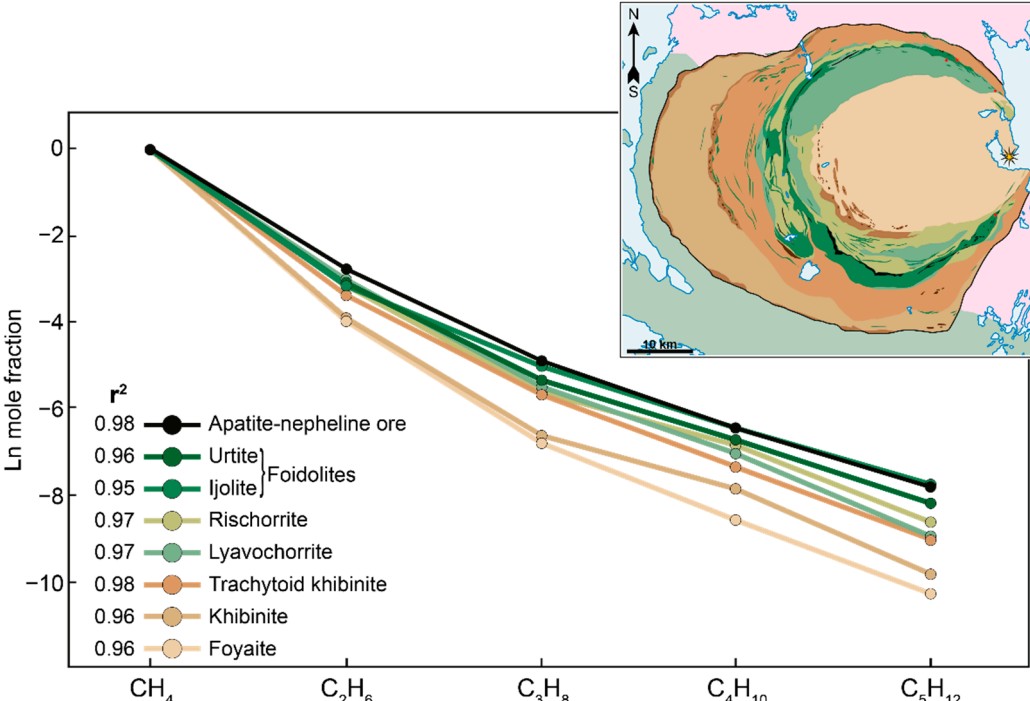

**Figure 3.** Median molecular weight distributions of hydrocarbon gases in the magmatic rocks and apatite–nepheline ore of the Khibiny massif. $r^2$—coefficients of determination to best-fit lines. The top right shows the geological scheme of the Khibiny massif (see also Figure 1). The colors of the lines on the graphs correspond to the colors of the rocks on the geological scheme of the massif for easier comparison.

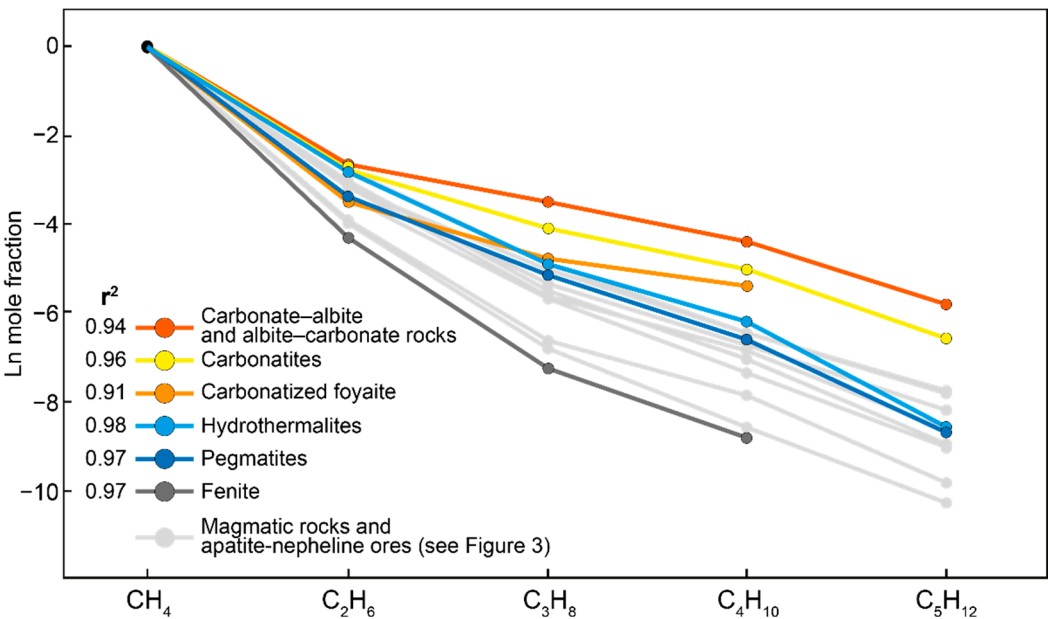

**Figure 4.** Median molecular weight distributions of hydrocarbon gases in the carbonatites, hydrothermalites, pegmatites, and other rocks of the Khibiny massif. $r^2$—coefficients of determination to best-fit lines.

## 5. Discussion

In accordance with the classical Anderson–Schulz–Flory distribution, a relatively steep slope of the linear dependence is considered as criterion of abiogenic origin of natural gases by Fischer–Tropch synthesis [2,30,36,67]. Molecular weight distribution of alkanes in the Khibiny rocks, except for the carbonatized foyaite, was consistent with the classical Anderson–Schulz–Flory distribution model. Indeed, plots of log $X_i$ versus $C_n$ (where X is concentration) are straight lines (Figures 3 and 4). By analogy with the experiments [68], some deviations of the Khibiny hydrocarbon gases distribution from straight linearity reflect the instability of thermodynamic conditions, multiple generations, and/or long duration of gas formation. The most noticeable deviations represent relatively elevated concentrations of butanes, especially in the rocks of the carbonatite complex and hydrothermalites (Figure 4).

However, the same molecular weight distribution character has been identified for hydrocarbon gases formed by polymerization. Indeed, the features of molecular weight distribution of alkanes in the Khibiny rocks confirm the abiogenic origin of the vast majority of the hydrocarbon gases. However, additional criteria are required in order to establish the exact mechanism for the formation of these gases. In the last two decades, post-magmatic, hydrothermal Fischer–Tropch type reactions are most often used to explain the origin of the hydrocarbon gases in peralkaline complexes [2,29,36], and the processes of polymerization of primary (presumably magmatic or even mantle), methane ($n\text{CH}_4 \rightarrow \text{C}_n\text{H}_{2n+2} + (n-1)\text{ H}_2$), and oxidation of hydrocarbon gases ($4\text{CH}_4 + \text{O}_2 \rightarrow 2\text{C}_2\text{H}_6 + 2\text{H}_2\text{O}$ and $3\text{CH}_4 + \text{O}_2 \rightarrow \text{C}_3\text{H}_8 + 2\text{H}_2\text{O}$) [22,69]. It is obvious that with the similarity of the composition, content, and sources of gas components in the considered alkaline complexes, the contribution of different sources and the nature of the gas phase evolution may differ not only in each massif, but also in the different rocks of the same massif.

Additional arguments for the formation of hydrocarbon gases in peralkaline complexes by Fischer–Tropch type reactions are follows: (a) the relationship between late-magmatic Fe-oxidation and the production of $\text{CH}_4$ [2]; (b) wide variations in $\delta^{13}\text{C}_{\text{CH}4}$ (−25.3/−3.2‰) [21,32]; (c) the assumed presence of reagents for Fischer–Tropch type reactions, namely $\text{CO}_2$ and $\text{H}_2$. According to thermodynamic calculations [33,64], $\text{CO}_2$ is the main (together with $\text{H}_2\text{O}$) component of the primary magmatic fluid. Molecular hydrogen

could be generated during the evolution of the magmatic fluid, as well as during the interaction of iron-containing minerals and aluminosilicates with water, for example, during the formation of aegirine, magnetite, cancrinite, and zeolites; (d) close spatial association of hydrocarbon fluid inclusions in nepheline with aegirine inclusions [3,30]; (e) immiscibility of $H_2O$ and $CH_4$ fluid inclusions at or below $CH_4$–$H_2O$ solvus [30,32].

The value of the coefficient of determination ($r^2$) for most molecular weight distribution graphs of alkanes from the Khibiny rocks is 0.96–0.98, decreasing to 0.94 in the rocks of the carbonatite complex and to 0.91 in the carbonatized foyaite, i.e., in all cases less than 0.99 (Figures 3 and 4). Hence, according to the criterion proposed by Etiope and Sherwood Lollar [13], the Khibiny hydrocarbon gases should be classified as predominantly, but not completely, abiogenic. If so, a relatively larger contribution of the biogenic component, e.g., due to organic matter coming along with meteoric waters, can be assumed in low-temperature carbonatization processes of nepheline syenites and formation of hydrothermal veins of zeolite–carbonate and albite–carbonate rocks. However, this criterion, like most others, does not seem to be universal and, outside the complex of other signs and characteristics, is not sufficient to assess the nature of hydrocarbons [38].

Earlier, using the example of the rocks from the Lovozero massif, as well as Khibiny and Lovozero minerals, it was shown that a decrease in the steepness of the molecular weight distributions graphs, as well as the $CH_4/C_2H_6$ ratio, reflects a descent in the temperature interval limit of the gas formation [70], post-magmatic processes, and trapping of fluid inclusions [27]. Therefore, we can assume a decrease in the temperature of the completion of gas formation and, probably, an increase in the time interval of this process in the following sequence of Khibiny rocks: foyaite and khibinite–trachytoid khibinite–rischorrite and lyavochorrite–foidolites and apatite–nepheline ores–carbonatites. This assumption is consistent with petrographic, geochemical, and mineral zoning that is symmetric with respect to the Main Ring [45,54–56]. The relatively late formation of the Khibiny carbonatites is evidenced, in particular, by recent data on the sulfur isotopic composition of sulfides [71]. In addition, occluded gases from the rocks of the carbonatite complex and hydrothermalites are distinguished by the lowest $nC_4H_{10}/iC_4H_{10}$ ratios, which may indicate the lowest gas formation temperatures [72].

Based on the slope of the molecular weight distribution, the formation of hydrocarbon gases was completed in fenite at the highest temperatures. A similar conclusion with respect to the fenite of the Lovozero massif follows from the nature of the distribution of methane and helium isotopes [26]. The formation temperatures of such fenite halos are estimated at 500–800 °C [73,74].

The slopes of the molecular weight distribution graphs of alkanes in other peralkaline massifs are close to those in different types of the Khibiny rocks (Figure 5). Hence, it is possible to assume similar conditions for gas formation in the Khibiny foyaite and foyaite of the deepest horizons of the Lovozero massif (lines 3 and 6), and in the Khibiny foidolites and rocks of the Ilímaussaq (lines 2 and 7–9), as well as in rocks of the carbonatite complex of the Khibiny massif and granite pegmatites of Strange Lake (lines 1 and 10–11). A probable decrease in the temperature of the fluid inclusion capture from bottom to top along the section of the Lovozero massif (lines 4–6) is consistent with other geochemical and mineralogical data [5,26]. The decrease in the slopes of the molecular weight distribution graphs in the series of Ilímaussaq rocks from naujaite to lujavrite corresponds to the sequence of their formation from the evolving initial melt [28]. It is assumed that in the Strange Lake alkaline–granite complex, hydrocarbon gases formed at temperatures below 400 °C as a result of Fischer–Tropsch type reactions [2] and/or oxidation and polymerization [69].

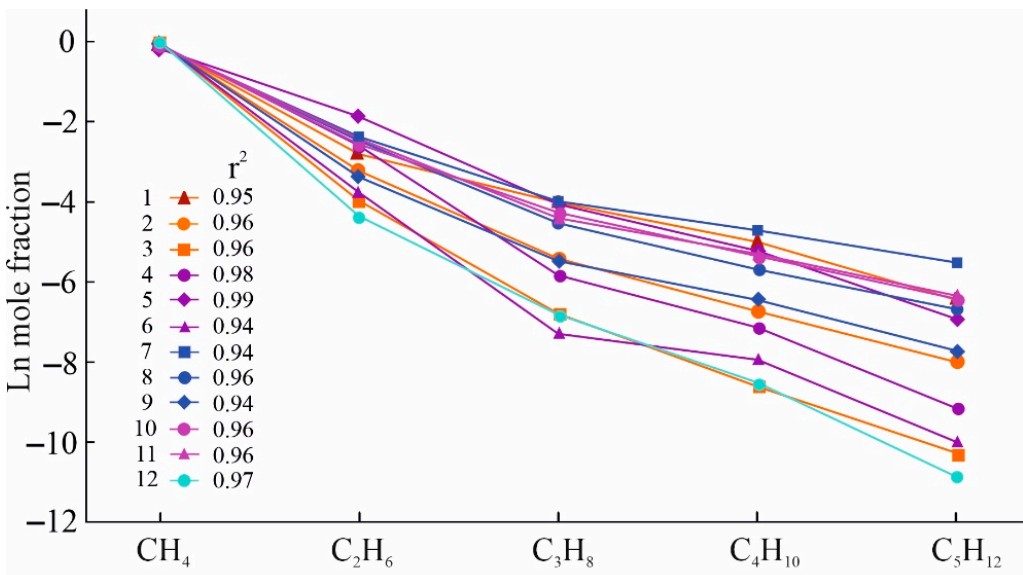

**Figure 5.** Molecular–weight distributions of alkanes in the rocks of peralkaline complexes. Khibiny massif: (1) Carbonatite complex, (2) foidolites, (3) foyaite; Lovozero massif [27]: urtite (4) and loparite malignite (5) of horizon II-4 (upper part of the Layered complex of the Lovozero massif), (6) foyaite of the series V (lower part of the Layered complex of the Lovozero massif); Ilímaussaq massif [31]: (7) lujavrite, (8) kakortokite, (9) naujaite; Strange Lake complex [2]: (10) "fresh" pegmatite, (11) altered pegmatite; Kiya–Shaltyr massif (unpublished data from A. Petersil'e): (12) urtite. $r^2$—coefficients of determination to best-fit lines.

Figure 6 compares the distribution of saturated hydrocarbons in natural gases of various origin. Note that all plots of the distribution of thermogenic alkanes have a smaller slope and lower values of determination coefficients compared to the distribution plots of alkanes from the Khibiny massif. The most high-temperature ones shown in this figure are, apparently, abiogenic hydrocarbons from a hot spring opened by a borehole in serpentinized ultramafic rocks of Hakuba Happo, Japan. It is assumed that these gases were formed by methane polymerization [40]. The distribution of hydrocarbon components in the gas from the source of the Olympic flame in Turkey, which is a mixture of organic and abiogenic ones [75], does not correspond to the classical Anderson–Schulz–Flory distribution. The molecular–weight distributions of gases obtained experimentally by Fischer–Tropch reactions at temperatures of 400 °C [76] turned out to be close to that of thermogenic gases. Figure 6 can serve as an illustration of the above-mentioned unreliability of the criterion for the formation of the hydrocarbons by Fischer–Tropch reactions, which is often considered to be the correspondence of their molecular–weight distributions to the classical one, provided that the slope of the linear dependence is relatively steep.

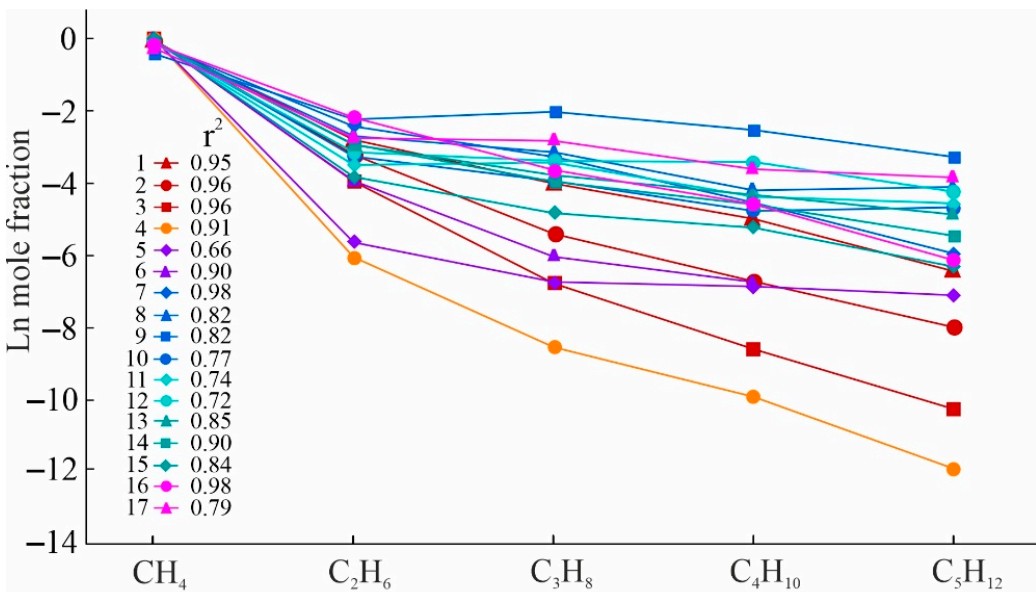

**Figure 6.** Molecular–weight distributions of hydrocarbons in gases of both geological settings and obtained during experiments. Khibiny massif: (1) Carbonatite complex, (2) foidolites, (3) foyaite; (4) Hakuba Happo hot spring, Japan [40]; (5) mixture of organic and abiogenic gas from the source of the Chimaera Olympic flame, Turkey [75]; (6) mixture of presumably abiogenic (ca. 80%) and organic gas in the Xujiaweizi Fault Depression, Songliao Basin, China [77]. Gas deposits of the (7) North Sea [78]; (8) West Siberia [79]; (9) North Siberia, The Medvezh'e deposit [80]; (10) Caucasus. (11,12) Gas of thermogenic and mixed (thermogenic plus bacterial) origin in the Thrace Basin, Turkey [81]. Oil-type (13), coal-type (14), and mixture (15) in the northern Dongpu Depression, Bohai Bay Basin, China [82]. (16,17) Gas synthesized in a laboratory by Fischer–Tropsch-type reactions at a temperature of 400 °C [76]. $r^2$—coefficient of determination.

Such features of the molecular–weight distributions of Khibiny hydrocarbons show an increase in the fraction of heavy alkanes as the gas formation temperature decreases, as well as the relatively higher concentrations of butanes and lower ethane, which may be evidence in favor of the Fischer–Tropch reactions. However, many facts and observations, and the results of some experiments and theoretical modeling, do not agree with the hypothesis of the hydrocarbon formation in nepheline–syenite massifs via Fischer–Tropch reactions.

One of the reasons for such assumptions is the absence of $CO_2$ in the primary fluid inclusions, which are almost purely methane in composition. The gas phase of melt inclusions is also mainly represented by methane [62]. Secondary $CO_2$ inclusions in the Khibiny massif have also so far been observed only in carbonatites. According to our data, there is no correlation between concentrations of $CO_2$ and $CH_4$, which would be expected in the case of predominant $CH_4$ generation by Fischer–Tropch reactions. When hydrocarbon is formed by Fischer–Tropch synthesis, along with the depletion of $^{13}C$, the enrichment of heavier hydrocarbon gases with deuterium should occur [83]. In the Khibiny massif, both $^{13}C$ and deuterium depletion are observed, which may be the result of low-temperature polymerization reactions [21,22].

The possible relatively early appearance of methane is evidenced by higher estimates of the $CH_4$–$H_2O$ isotopic equilibrium temperature compared to $CH_4$–$H_2$ and $H_2$–$H_2O$ in the Lovozero gases [84]. Synthesis of hydrocarbons by polymerization of methane is also possible under hydrothermal conditions [85], whereas Fischer–Tropch type reactions with an intensive water circulation and a low gas/water ratio are relatively ineffective [39,86].

It is known that in most geological and biological systems, $CH_4$ is depleted in $^{13}C$ relative to coexisting $CO_2$. In the Khibiny gases, the opposite situation, which is quite rare in nature, is observed. Thus, according to Beeskow et al. [32], the average (from eight analyses) $\delta^{13}C_{CH_4}$ and $\delta^{13}C_{CO_2}$ concentrations are −11.9‰ and −14.7‰, respectively. In urtite

and apatite–nepheline ore, $\delta^{13}C_{CH4}$ and $\delta^{13}C_{CO2}$ concentrations are −10.3‰ and −16.3‰, respectively [21]. Such a situation is explained by the formation of small $CO_2$ amounts due to the late abiogenic oxidation of $CH_4$ as a result of kinetic fractionation. In other processes of methane oxidation, the formation of carbon dioxide can occur by the following reaction: $CH_4 + 2O_2 \rightarrow CO_2 + 2H_2O$. In the case of predominant oxygen consumption, its homologues can be generated, for example, by the following reactions: $4CH_4 + O_2 \rightarrow 2C_2H_6 + 2H_2O$ and $3CH_4 + O_2 \rightarrow C_3H_8 + 2H_2O$ [69].

These data allow us to assume a polygenic, overwhelmingly, if not completely, abiogenic origin, non-single-stage formation, and transformation of the Khibiny hydrocarbon gases, which occurred at different stages of mineral formation. In general, in the Khibiny massif, the proportion of relatively high-temperature gases decreases towards the Main Ring in the following sequence: foyaite and khibinite–trachytoid khibinite–rischorrite, lyavochorrite–foidolites, and apatite–nepheline ores–carbonatites. In the same sequence, there is an increase in the proportion of heavy hydrocarbons of hydrocarbon gases, and the increasing role of oxidation and condensation reactions in the transformation of hydrocarbons occurs.

As in Lovozero rocks [26], the presence of an insignificant fraction of mantle $CH_4$ or, at least, its initial carbon, cannot yet be excluded in Khibiny. At the magmatic stage of the formation of the Khibiny massif, methane in the fluid could appear during the crystallization of aegirine and alkaline amphiboles [24]. The interaction of water and previously formed graphite could be considered as another possible way of the hydrocarbon gas generation [33]. In the course of the further system's evolution (temperature decrease, nonuniform dilution of the magmatic fluid by circulating paleometeoric waters, etc.), a complex multi-stage process of hydrocarbon formation, which included reactions of polymerization, Fischer–Tropsch type reactions, oxidation, dehydrogenation, and condensation, was apparently launched. Such transformations could occur in a fairly wide field of changing thermodynamic parameters [87,88]. Some additional factors could also affect the composition of the occluded gases. These are, for example, the formation of $H_2$ and heavier hydrocarbon gases due to the radiolysis of water and methane [89] and the $H_2$ diffusion both inside and out of the inclusion that weakens with a temperature decrease [90].

The established direction of the hydrocarbon gases compositional evolution is consistent with the distribution of condensed organic matter observed mainly in late hydrothermal mineral associations [91–93]. Such hydrothermal transformations of hydrocarbons were obviously facilitated by the dilution of residual magmatic fluids by infiltration surface waters, which was established from the isotopic composition of oxygen [94]. These waters could also bring a small amount of biogenic organic matter. This can explain the insignificant deviations of the molecular weight distribution of the Khibiny alkanes from the classical Anderson–Schulz–Flory distribution and the decrease in the value of the coefficient of determination.

For a better understanding of the sources, conditions, and mechanisms of the formation and evolution of hydrogen–hydrocarbon gases in the rocks and minerals of the nepheline–syenite massifs, it is necessary to combine various approaches and research methods. These are such methods as gas chromatography and mass spectrometry, thermobarometry of fluid inclusions, isotopic analysis, and detailed mineralogical and geochemical observations.

## 6. Conclusions

1.　The molecular weight distribution of occluded hydrocarbon gases in the Khibiny massif corresponds to the classical Anderson–Schulz–Flory distribution. In addition, the slopes of the linear relationships are relatively steep. This indicates a predominantly abiogenic origin of the occluded gases of the Khibiny massif. At the same time, a small proportion of biogenic hydrocarbons is present and is associated with the influence of meteoric waters.

2. The mechanism of formation of hydrocarbons remains debatable. The most probable ways of their formation are Fischer–Tropsch reactions ($nCO_2 + (3n + 1)H_2 \rightarrow C_nH_{2n+2} + 2nH_2O$), processes of polymerization of primary methane ($nCH_4 \rightarrow C_nH_{2n+2} + (n - 1) H_2$), and oxidation of hydrocarbon gases ($4CH_4 + O_2 \rightarrow 2C_2H_6 + 2H_2O$).

3. In the Khibiny massif, the proportion of relatively high-temperature gases decreases towards the Main foidolite Ring in the following sequence: foyaite and khibinite–trachytoid khibinite–rischorrite and lyavochorrite–foidolites and apatite–nepheline ores. In the same sequence, there is an increase in the proportion of heavy hydrocarbons of hydrocarbon gases, and the increasing role of oxidation and condensation reactions in the transformation of hydrocarbons occurs.

4. The pattern of the molecular weight distribution of hydrocarbon gases can serve as an indicator of the conditions and mechanism of their formation, but only in combination with other signs and criteria.

**Author Contributions:** Conceptualization, V.A.N.; formal analysis, V.A.N.; investigation, V.A.N.; data curation, V.A.N., V.V.P. and O.D.M.; writing—original draft preparation, V.A.N.; writing—review and editing, V.V.P. and J.A.M.; visualization, O.D.M. and J.A.M. All authors have read and agreed to the published version of the manuscript.

**Funding:** This research was funded by the Russian Science Foundation, project no. 21-47-09010.

**Data Availability Statement:** Not applicable.

**Acknowledgments:** We are grateful to reviewers from MDPI who helped us improve the presentation of our results. The authors thank Alexandra Rybnikova for editing the English version of the manuscript.

**Conflicts of Interest:** The authors declare no conflict of interest.

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
