# Peer review of "The Molecular Weight Distribution of Occluded Hydrocarbon Gases in the Khibiny Nepheline–Syenite Massif (Kola Peninsula, NW Russia) in Relation to the Problem of Their Origin"

_geosciences, doi:10.3390/geosciences12110416_

Round 1
Reviewer 1 Report
The manuscript is about an up-to-date and widely discussed subject concerning the “origin of hydrogen–hydrocarbon gases present in concentrations unusually high for some magmatic complexes” and I think it will interest the readers of the Geosciences Journal.
Even though this subject is not my specialization, the manuscript is suitable for reading and I understand and agree with all the propositions and the related results and discussions.
Although I am not a native speaker, I do not have problems to understand the English. It is very clear and I do not see problems. But maybe as I am not a native speaker it is possible to suggest that the English be revised and improved in a manner more suitable for publication.
It is an interesting manuscript and the main objective is reached.
But I think that the manuscript must give more information specially about location and geology concerning the studied area. The figure 1 could be improved and give more details because is difficult to an external reader identify the area.
Results, discussions, and conclusions are excellent.
Author Response
The manuscript is about an up-to-date and widely discussed subject concerning the “origin of hydrogen–hydrocarbon gases present in concentrations unusually high for some magmatic complexes” and I think it will interest the readers of the Geosciences Journal. Even though this subject is not my specialization, the manuscript is suitable for reading and I understand and agree with all the propositions and the related results and discussions. Although I am not a native speaker, I do not have problems to understand the English. It is very clear and I do not see problems. But maybe as I am not a native speaker it is possible to suggest that the English be revised and improved in a manner more suitable for publication. It is an interesting manuscript and the main objective is reached. But I think that the manuscript must give more information specially about location and geology concerning the studied area. The figure 1 could be improved and give more details because is difficult to an external reader identify the area. Results, discussions, and conclusions are excellent.
Response. We are grateful for the high appreciation of our work and hope that the article will be of interest to readers. We have done significant revision and editing of the English. We also changed Figure 1 (geological scheme of the Khibiny massif) and made the description of the geology of the massif more detailed (lines 92-133).
Reviewer 2 Report
Hi all,
a really interesting paper with novel data are hiding in this submission. However a significant re-write may be required to unlock the information. This paper needs a good internal review by your co-authors and colleagues to help provide important clarifications to the work.
I have attached a partial edit of the paper, but I was frustrated by the lack of clarification of points in the text. I was unable to work out why references were cited - I do not expect to go read over 100 papers to understand your paper.
Context around points being made were absent, so I could not work out what the authors were trying to convey. This means that the purpose and value of the work done is lost to the reader.
My recommendation is that you revise your manuscript and consider that if the reader has to ask "who, what, why, where or when" for a sentence, then it probably requires more clarification.
My other concern is the lack of detail regarding the methods used. There is a suggestion that this work is a compilation of either data or samples that have been tested. I'm not 100% clear about this. If more than one method was used, this has not been clarified. Nor have risks and artefacts caused by the method(s) used discussed. This is important to understand the accuracy of the represented results.
I recommend that the authors have another go at revising this paper as it seems to me that it is potentially significant work and adds value to the topics of abiogenic gas and naturally occurring hydrogen (a hot topic right now!).
One final comment. There was extensive use of really unusual acronyms that popped up - not everything needs a TLA (three letter acronym ;-)

Author Response
Hi all,
a really interesting paper with novel data are hiding in this submission. However a significant re-write may be required to unlock the information.
Point 1. This paper needs a good internal review by your co-authors and colleagues to help provide important clarifications to the work. I have attached a partial edit of the paper, but I was frustrated by the lack of clarification of points in the text. I was unable to work out why references were cited - I do not expect to go read over 100 papers to understand your paper. Context around points being made were absent, so I could not work out what the authors were trying to convey. This means that the purpose and value of the work done is lost to the reader. My recommendation is that you revise your manuscript and consider that if the reader has to ask "who, what, why, where or when" for a sentence, then it probably requires more clarification.
Response 1. The manuscript has been substantially revised and rewritten. Some parts of the manuscript, such as abstract, geological background, conclusions, have been completely changed. In addition, Figures 1 and 3 have been modified. Extensive editing of the English was also done.
Point 2. My other concern is the lack of detail regarding the methods used. There is a suggestion that this work is a compilation of either data or samples that have been tested. I'm not 100% clear about this. If more than one method was used, this has not been clarified. Nor have risks and artefacts caused by the method(s) used discussed. This is important to understand the accuracy of the represented results. I recommend that the authors have another go at revising this paper as it seems to me that it is potentially significant work and adds value to the topics of abiogenic gas and naturally occurring hydrogen (a hot topic right now!).
Response 2. The manuscript has been substantially revised and rewritten. The description of the methodology has been significantly changed (lines 204-247).
Point 3. One final comment. There was extensive use of really unusual acronyms that popped up - not everything needs a TLA (three letter acronym ;-)
Response 3. All acronyms have been changed to normal words.

Reviewer 3 Report
The reviewed manuscript is dealing with the molecular weight distribution of occluded hydrocarbon gases in the Khibiny nepheline–syenite massif (Kola Peninsula, NW Russia) in relation to the problem of their origin. The authors touch on the important problem of using MWD features of occluded saturated HCGs for origin identification. In general, the manuscript is well-structured and interesting. The manuscript can be accepted after a minor revision.
1. The abstract is not self-reliant and not complete and should be improved.
2. Lines 42:51 and 55:60; It will be better if the authors rearranged the cited references in these paragraphs. For example, the provision of sustainable energy supply at low ecological and economic costs [00], the abiogenic synthesis of organic molecules [00], the development of the deep biosphere [00], …., …
3. Figure 1; This figure should be improved by adding a coordinate and a clearer scale. Consider improving the quality and the size of the base maps showing the exact location of the Khibiny massif.
4. Line 217; Nivin in his thesis [24] change to Nivin [24].
5. Table 1A and 1B; Please change to Table 1 and Table 2.
6. Line 237; Please provide specific information about the model of the used chromatograph.
7. Line 313; proposed in [32] change to proposed by Etiope and Sherwood Lollar [32].
8. Line 440; according to [28] change to according to Beeskow et al. [28].
9. The conclusions should be shortened and presented in terms of the characteristic findings of the study.
Author Response
The reviewed manuscript is dealing with the molecular weight distribution of occluded hydrocarbon gases in the Khibiny nepheline–syenite massif (Kola Peninsula, NW Russia) in relation to the problem of their origin. The authors touch on the important problem of using MWD features of occluded saturated HCGs for origin identification. In general, the manuscript is well-structured and interesting. The manuscript can be accepted after a minor revision.
We are grateful for the high appreciation of our work and hope that the article will be of interest to readers.
Point 1. The abstract is not self-reliant and not complete and should be improved.
Response 1. The abstract has been rewritten (lines 10-25). The abstract now gives give a pertinent overview of the work.
Point 2. Lines 42:51 and 55:60; It will be better if the authors rearranged the cited references in these
paragraphs. For example, the provision of sustainable energy supply at low ecological and economic
costs [00], the abiogenic synthesis of organic molecules [00], the development of the deep biosphere [00],
…., …
Response 2. The cited references have been rearranged.
Point 3. Figure 1; This figure should be improved by adding a coordinate and a clearer scale. Consider
improving the quality and the size of the base maps showing the exact location of the Khibiny massif.
Response 3. Figure 1 has been modified.
Point 4. Line 217; Nivin in his thesis [24] change to Nivin [24].
Response 4. Corrected.
Point 5. Table 1A and 1B; Please change to Table 1 and Table 2.
Response 5. Corrected.
Point 6. Line 237; Please provide specific information about the model of the used chromatograph.
Response 6. Information about the chromatograph model has been added (lines 237-238).
Point 7. Line 313; proposed in [32] change to proposed by Etiope and Sherwood Lollar [32].
Response 7. Corrected (lines 323-324).
Point 8. Line 440; according to [28] change to according to Beeskow et al. [28].
Response 8. Corrected (line 423).
Point 9. The conclusions should be shortened and presented in terms of the characteristic findings of the study.
Response 9. The conclusions have been shortened and now presented as four small paragraphs (lines 471-490).
Round 2
Reviewer 2 Report
Hi Authors. Wow! Great work on the edits. This has transformed the paper and made it much more accessible. Thank you for taking on board my concerns and turning it around.
I had a couple of minor grammatical errors that I've flagged on the document. It was a little challenging to read with the track changes included, but it was great that you left them in place to demonstrate the large re-write that was undertaken.
Thank you.

Author Response
We are grateful for the high appreciation of our work and hope that the article will be of interest to readers. We have made all corrections to the text in accordance with the requirements of the reviewer.